# Development of a risk score for prediction of poor treatment outcomes among patients with multidrug-resistant tuberculosis

Kefyalew Addis Alene[1,2,3]*, Kerri Viney[1,4], Darren J. Gray[1], Emma S. McBryde[5], Zuhui Xu[6], Archie C. A. Clements[3]

**1** Research School of Population Health, College of Health and Medicine, The Australian National University, Canberra, Australian Capital Territory, Australia, **2** Institute of Public Health, College of Medicine and Health Sciences, University of Gondar, Gondar, Ethiopia, **3** Faculty of Health Sciences, Curtin University, Perth, Western Australia, Australia, **4** Department of Public Health Sciences, Karolinska Institutet, Stockholm, Sweden, **5** Australian Institute of Tropical Health and Medicine, James Cook University, Townsville, Queensland, Australia, **6** Department of Tuberculosis Control, Tuberculosis Control Institute of Hunan Province, Changsha city, Hunan Province, China

* kefyalew.alene@telethonkids.org.au

**Data Availability Statement:** All relevant data are within the paper and its Supporting Information files.

## Abstract

### Background

Treatment outcomes among patients treated for multidrug-resistant tuberculosis (MDR-TB) are often sub-optimal. Therefore, the early prediction of poor treatment outcomes may be useful in patient care, especially for clinicians when they have the ability to make treatment decisions or offer counselling or additional support to patients. The aim of this study was to develop a simple clinical risk score to predict poor treatment outcomes in patients with MDR-TB, using routinely collected data from two large countries in geographically distinct regions.

### Methods

We used MDR-TB data collected from Hunan Chest Hospital, China and Gondar University Hospital, Ethiopia. The data were divided into derivation (n = 343; 60%) and validation groups (n = 227; 40%). A poor treatment outcome was defined as treatment failure, lost to follow up or death. A risk score for poor treatment outcomes was derived using a Cox proportional hazard model in the derivation group. The model was then validated in the validation group.

### Results

The overall rate of poor treatment outcome was 39.5% (n = 225); 37.9% (n = 86) in the derivation group and 40.5% (n = 139) in the validation group. Three variables were identified as predictors of poor treatment outcomes, and each was assigned a number of points proportional to its regression coefficient. These predictors and their points were: 1) history of taking second-line TB treatment (2 points), 2) resistance to any fluoroquinolones (3 points), and 3) smear did not convert from positive to negative at two months (4 points). We summed these

**Funding:** Kerri Viney is funded by the National Health and Medical Research Council (NHMRC) through Sidney Sax Early Career Fellowship, and Archie Clements was funded by a NHMRC Senior Research Fellowship. The funder had no role in study design, data collection and analysis, interpretation of data, decision to publish, or preparation of the manuscript.

**Competing interests:** The authors have declared that no competing interests exist.

points to calculate the risk score for each patient; three risk groups were defined: low risk (0 to 2 points), medium risk (3 to 5 points), and high risk (6 to 9 points). In the derivation group, poor treatment outcomes were reported for these three groups as 14%, 27%, and 71%, respectively. The area under the receiver operating characteristic curve for the point system in the derivation group was 0.69 (95% CI 0.60 to 0.77) and was similar to that in the validation group (0.67; 95% CI 0.56 to 0.78; p = 0.82).

## Conclusion

History of second-line TB treatment, resistance to any fluoroquinolones, and smear non-conversion at two months can be used to estimate the risk of poor treatment outcome in patients with MDR-TB with a moderate degree of accuracy (AUROC = 0.69).

## Introduction

Multidrug-resistant tuberculosis (MDR-TB) is an emerging public health problem that affected an estimated 457 000 people in 2017 [1]. The treatment of MDR-TB is lengthy (nine to 24 months), complicated by potentially severe adverse effects, and is more costly than treatment of drug susceptible tuberculosis (TB) [2, 3]. In addition, treatment outcomes are sub-optimal [4–6], with only 54% of MDR-TB patients successfully treated globally [7]. A range of clinical and demographic factors have been shown to be associated with poor treatment outcomes, including drug resistance pattern, history of TB treatment, and smear result at the commencement of treatment [5, 6, 8–13]. Developing a clinical risk score using readily available predictors to identify individuals at increased risk of poor treatment outcomes could be a useful aid in clinical decision making. Risk scores have been developed to predict the risk of TB treatment outcomes (for drug susceptible TB) [14], severe dengue [15], coronary heart disease [16, 17], paediatric mortality [18], and pulmonary embolism [19]. However, risk scores have not yet been developed to predict treatment outcomes among patients with MDR-TB. Therefore, the aim of this study was to develop a clinical risk score to predict the risk of poor treatment outcome (i.e. treatment failure, lost to follow up or death) in patients with MDR-TB, using routinely collected data from two large countries that have a high MDR-TB burden and are geographically, economically and epidemiologically distinct.

## Methods

### Study design and settings

We conducted a retrospective cohort study using data collected from the MDR-TB treatment centres at Gondar University Hospital, Ethiopia, and Hunan Chest Hospital, China. According to the World Bank classification, Ethiopia is categorized as a low-income country, whereas China is categorized as an upper-middle-income country [20]. However, both countries are categorized by World Health Organization (WHO) as high TB and MDR-TB burden countries [21]. Hunan Chest Hospital is located in Changsha, the capital city of Hunan province, and Gondar University Hospital is located in northwest Ethiopia. The Hunan Chest Hospital and the Gondar University Hospital established MDR-TB treatment centers in 2011 and 2010, respectively. These hospitals provide comprehensive diagnostic and treatment services for all people with presumptive drug resistant TB who live in their catchment area.

## MDR-TB diagnosis

At Gondar University Hospital, from patients suspected of having DR-TB, sputum specimens are sent to the national or regional laboratory for culture, and drug resistance is determined by a line probe assay, Xpert® MTB/RIF or conventional drug susceptibility testing (DST); whereas at the Hunan Chest Hospital, culture and DST are performed in the hospital's laboratory. In addition, sputum specimens from all culture-positive TB patients from throughout the province are referred to the Hunan Chest Hospital for DST. In the hospital, phenotypic DST based on solid and liquid culture techniques, and molecular methods using line probe assays as well as Xpert® MTB/RIF are performed. Both in Hunan Chest Hospital and Gondar University Hospital, smear microscopy with Ziehl-Neelsen staining and fluorescence microscopy are used for the diagnosis and monitoring of TB.

## MDR-TB treatment

At Hunan Chest Hospital, patients with MDR-TB are treated with an individualized treatment regimen containing at least four drugs, based on their DST results and history of previous TB treatment. The regimen usually includes an injectable agent (i.e. Kanamycin, Amikacin or Capreomycin), a Fluoroquinolone (i.e. Levofloxacin, Ofloxacin or Moxifloxacin), Para-amino-salicylic acid, Prothionamide, Pyrazinamide, Clarithromycin, Ethambutol, or Cycloserine. The duration of treatment is approximately 24 months; and the injectable drugs are used for a minimum of six months. Patients are admitted to the hospital for one to two months during the intensive phase and, while hospitalised, receive directly observed therapy (DOT) by trained medical staff. During this time, patients also receive psychological support and counselling from hospital nurses. When the patients are medically fit, they are treated as out-patients. They receive support from trained family members or from trained supervisors in the community and return to the hospital once a month for a drug refill. As part of routine care, sputum microscopy and cultures are performed monthly for the first six months, and thereafter every other month until the end of treatment.

At Gondar University Hospital, MDR-TB patients receive a standardized regimen of first and second line TB drugs that consists of an eight-month intensive phase with a combination of Pyrazinamide, Capreomycin, Levofloxacin, Prothionamide or Ethionamide and Cycloserine, a twelve-month continuation phase with a combination of Pyrazinamide, Levofloxacin, Prothionamide or Ethionamide, and Cycloserine [22]. However, certain groups of MDR-TB patients cannot receive the standardized regimen. These groups include: pregnant women, children, patients with co-morbidities such as chronic renal dysfunction, HIV or liver disease, patients who report household contact with other rifampicin resistant (RR) or MDR-TB patients, and patients who have a history of prior exposure to second-line TB drugs [22]. These group of patients require either a modification of the regimen or dose adjustment [22].

## Data source

This study included all patients who were diagnosed with MDR- TB in Hunan Chest Hospital from 2011–2014, and in Gondar University Hospital from 2010–2014. Patients who did not start MDR-TB treatment and patients whose treatment result were not recorded were excluded from the study. Demographic variables such as age, sex and occupation, and clinical variables such as history of second-line TB treatment, resistance to ethambutol, resistance to injectable TB drugs, resistance to fluoroquinolones, culture and smear results (at baseline and at follow up), and treatment outcomes were obtained from the patients' medical records (S1 File). To develop and validate the risk score, the data obtained from the two hospitals were merged

together, and divided into a derivation group (60%) and validation group (40%) using a random number generator in STATA version 15 [23].

## Definitions

Multidrug-resistant TB was defined as TB that is resistant to at least isoniazid and rifampicin [24]. A poor treatment outcome was defined as the sum of the treatment outcomes: treatment failure, lost to follow up or death. Treatment failure was defined as treatment terminated or a need for permanent regimen change of at least two TB drugs due to an adverse drug reaction, or lack of culture conversion by the end of the intensive phase, or bacteriological reversion in the continuation phase after conversion to negative after the intensive phase, or evidence of additional acquired resistance to fluoroquinolones or second-line injectable drugs [24, 25]. Lost to follow-up was defined as a patient whose treatment was interrupted for two consecutive months or more [24, 25]. Death was defined as patients who died for any reason during the course of treatment [24, 25]. Culture conversion (from positive to negative) was defined as two consecutive negative sputum cultures taken at least 30 days apart following an initial positive culture [25]. Similarly, smear conversion (from positive to negative) was defined as two consecutive negative sputum smears taken at least 30 days apart following an initial positive sputum smear [25].

## Statistical analysis

Categorical variables were presented as numbers and percentages, and continuous variables were presented as means and standard deviations. The rate of poor treatment outcome was calculated as the proportion of patients who had treatment failure lost to follow up or death during the follow-up period among those who had started MDR-TB treatment.

The risk score was developed from the derivation group, using previously established method [26]. We first performed univariable cox-proportional hazard models using demographic and clinical variables. We selected variables which had a p value <0.2 in the univariable analysis and checked for the presence of collinearity between the selected variables, using Pearson's correlation coefficient. A stepwise multivariable cox-proportional hazard model was used to select the independent predictors (with a p-value <0.05), and to estimate their regression coefficients (β). Since the Pearson's correlation coefficient showed that smear and culture conversion status were correlated, we selected smear conversion and removed culture conversion from the model. As a sensitivity analysis, we run two different models that contain culture conversion and smear conversion. Because these two models provided similar results, we preferred only the model that contain smear conversion. We selected smear conversion because it is less expensive, widely available, and more convenient than culture conversion to predict the treatment outcomes of patients at peripheral and lower-resource settings.

A score for each of the identified variables was calculated based on their regression coefficients. The regression coefficients (β) for the selected risk factors were estimated from the multivariable Cox proportional hazards model. The score for these variables was calculated by dividing the coefficient of the variable by the lowest β value in the model, multiplied by a constant, and rounded to the nearest integer. The constant was determined based on the Framingham study, in terms of the increase in risk of death and treatment failure associated with an increase in age. The constant for our study model was set at two. A risk score was finally calculated for each patient by summing up the points of all risk factors, and the population was divided into tertiles: patients at low risk, patients at medium risk, and patients at high risk for poor treatment outcome.

To evaluate the performance of the scoring systems, we assessed the discrimination (i.e. the ability of the scoring system to distinguish between patients with poor treatment outcomes and patients with successful treatment outcomes) and calibration (i.e. the level of agreement between predicted probabilities and observed outcomes) power of the score.

The receiver operating characteristic curve (ROC) was plotted and the area under the curve (AUC) was calculated to measure and compare the discriminatory power of the scoring system in the derivation and validation group. The 95% CI for the ROC curve was calculated by STAT, using "*diagt*" command.

The expected number of poor treatment outcome as predicted by the scoring system was compared with the observed number of poor treatment outcome in each group. A value of $P > 0.05$ was accepted as good calibration. Calibration was shown graphically by plotting the observed and predicted poor treatment outcomes, grouped according to quartile of predicted probabilities. All the analyses were performed using Stata Statistical Software: Release 14.0 (College Station, TX: Stata Corporation LP).

### Ethics approval and consent to participate

Ethics approval was obtained from the Australian National University Human Research Ethics Committee (protocol number 2016/218) and from the Institutional Review Board of the University of Gondar. Permission was granted to access the secondary data from Tuberculosis Control Institute of Hunan Province, and this was documented in a letter. All data were fully anonymized before we accessed them, and the study was conducted in collaboration with researchers from Gondar University and Hunan Chest Hospital.

## Results

### Baseline characteristics

A total of 570 bacteriologically confirmed MDR-TB patients were included in the study (Fig 1). The baseline demographic and clinical characteristics of the patients are presented in Table 1. Of 570 patients, 343 (60%) randomly selected patients were assigned in the deviation group, and the remaining 227 (40%) patients were assigned in the validation group. The characteristics of the study participants in Hunan Chest Hospital, China and University of Gondar, Ethiopia are presented in Table 2.

### Predictors of poor treatment outcomes

During the follow up period, the overall rate of poor treatment outcomes was 39.5% (n = 225); 37.9% (n = 86) in the derivation group and 40.5% (n = 139) in the validation group (Table 3). Table 4 shows the crude hazard ratios from the univariable Cox proportional hazard model in the derivation group in patients with MDR-TB. Variables with *p*-value less than 0.2 in the univariable Cox proportional hazard model were selected for the multivariable Cox proportional hazard model.

### Risk score derivation and validation

In the final multivariable Cox proportional hazard model, three variables were identified as independent predictors of poor treatment outcomes (with a p-value <0.05), and each was assigned a number of points, proportional to its regression coefficient. These predictors and their points were: 1) history of taking second-line TB treatment (2 points), 2) resistance to any fluoroquinolones (3 points), and 3) smear did not convert from positive to negative at two months (4 points) (Table 5). The score for these variables was calculated by dividing the

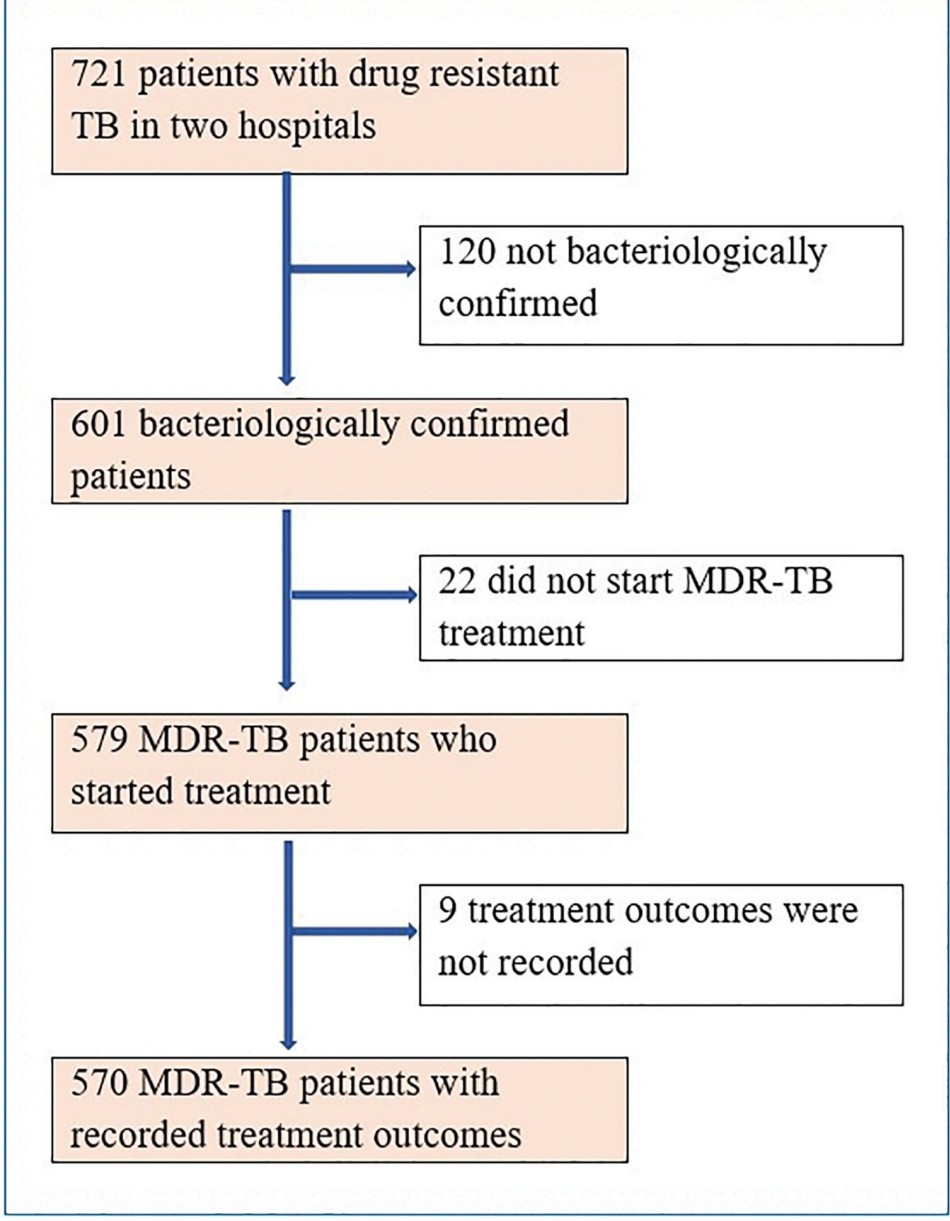

**Fig 1. Flowchart of eligible participants for our study on the development of a risk score for prediction of poor treatment outcomes among patients with multidrug-resistant tuberculosis, and reasons for exclusion.**

coefficient of the variable by 0.5 (the lowest $\beta$ value, corresponding to history of taking second-line TB treatment), multiplied by a constant (2), and rounded to the nearest integer. We summed these points to calculate the risk score for each patient, and three risk groups were defined: low risk (0 to 2 points), medium risk (3 to 5 points), and high risk (6 to 9 points).

Based on the risk score classification of the patients, in the derivation group, 230 (67%) patients were assigned to the low-risk group, 89 (26%) patients to medium risk group, and 24 (7%) patients to the high-risk group. The result was similar for the validation group: 163 (72%) patients were in the low-risk group, 46 (20%) in the medium-risk group, and 18 (8%) in the high-risk group.

**Table 1. Baseline demographic and clinical characteristics of patients with multidrug resistant tuberculosis in Hunan Chest Hospital, China and University of Gondar, Ethiopia, 2010–2014.**

| Variables | Cured | Treatment completed | Death | Treatment failure | Lost to follow up | Total |
|---|---|---|---|---|---|---|
| | N = 325; n (%) | N = 20; n (%) | N = 18; n (%) | N = 66; n (%) | N = 141; n (%) | N = 570; n (%) |
| **Age < 35 years** | 158 (48.6) | 9 (45) | 5 (27.8) | 23 (34.8) | 49 (34.8) | 244 (42.8) |
| Male sex | 211 (64.9) | 17 (85.0) | 16 (88.9) | 45 (68.2) | 107 (75.9) | 396 (69.4) |
| **Occupation** | | | | | | |
| Employed | 35 (10.8) | 6 (30.0) | 3 (16.8) | 3 (4.5) | 7 (5.0) | 54 (9.5) |
| Farmer | 224 (68.9) | 9 (45.0) | 11 (61.1) | 50 (75.8) | 108 (76.6) | 402 (70.5) |
| Unemployed | 22 (6.8) | 0 | 2 (11.1) | 1 (1.52) | 3 (2.13) | 28 (4.9) |
| Daily labourer | 18 (5.5) | 2 (10.0) | 1 (5.6) | 1 (1.5) | 9 (6.4) | 31 (5.4) |
| Others | 11 (3.4) | 0 | 1 (5.6) | 6 (9.1) | 4 (2.8) | 22 (3.9) |
| Unknown | 15 (4.6) | 3 (15) | 0 | 5 (7.6) | 10 (7.1) | 33 (5.8) |
| **Year of enrolment** | | | | | | |
| 2010 | 2 (0.6) | 1 (5.0) | 1 (5.6) | 0 | 0 | 4 (0.7) |
| 2011 | 10 (3.1) | 2 (10.0) | 1 (5.6) | 0 | 0 | 13 (2.3) |
| 2012 | 80 (24.6) | 6 (30.0) | 8 (44.4) | 19 (28.8) | 26 (18.4) | 139 (24.4) |
| 2013 | 107 (32.9) | 9 (45.0) | 4 (22.2) | 31 (47.0) | 43 (30.5) | 194 (34.0) |
| 2014 | 126 (38.8) | 2 (10.0) | 4 (22.2) | 16 (24.2) | 72 (51.1) | 220 (38.6) |
| **Study settings** | | | | | | |
| Hunan Chest Hospital, China | 256 (78.8) | 14 (70.0) | 13 (72.2) | 63 (95.4) | 132 (93.6) | 478 (83.9) |
| Gondar University Hospital, Ethiopia | 69 (21.2) | 6 (30.0) | 5 (27.8) | 3 (4.6) | 9 (6.4) | 92 (16.1) |
| **History of TB treatment** | 306 (94.1) | 19 (95.0) | 18 (100) | 65 (98.5) | 129 (91.5) | 537 (94.2) |
| **History of 2nd line TB treatment** | 70 (21.5) | 3 (15.0) | 8 (44.4) | 26 (39.4) | 31 (22.0) | 138 (24.2) |
| **Resistant to Ethambutol** | 100 (30.8) | 6 (30.0) | 9 (50.0) | 24 (36.4) | 45 (31.9) | 184 (32.3) |
| **Resistant to an injectable TB drugs** | 177 (54.7) | 12 (60.0) | 10 (55.6) | 42 (63.6) | 65 (46.1) | 306 (53.7) |
| **Resistant to any fluoroquinolone** | 17 (5.2) | 0 | 2 (11.1) | 15 (22.7) | 12 (8.5) | 46 (8.1) |

**Table 2. The demographic and clinical characteristics of the patients by study sites (Hunan Chest Hospital, China and University of Gondar, Ethiopia), 2010–2014.**

| Risk factors | All Patients (N = 570) | China (N = 478) | Ethiopia (N = 92) |
|---|---|---|---|
| | Mean (SD*) | Mean (SD*) | Mean (SD*) |
| Age (years) | 38.0 ± 12.5 | 40.31± 2.8 | 32.44± 12.2 |
| | Number (%) | Number (%) | Number (%) |
| Male sex | 289 (67.4) | 338 (70.7) | 58 (63.0) |
| Farmer or daily labourer | 341 (79.5) | 395 (82.6) | 56 (60.9) |
| History of TB treatment | 408 (95.1) | 446 (93.3) | 91 (98.9) |
| History of 2nd line TB drug treatment | 107 (24.9) | 137 (28.6) | 1 (1.1) |
| Resistance to ethambutol | 139 (32.4) | 165 (34.5) | 19 (20.6) |
| Resistance to any injectable TB drugs | 243 (56.6) | 289 (60.5) | 20 (21.7) |
| Resistance to any fluoroquinolones | 34 (7.93) | 46 (9.6) | 0 (0.0) |
| Culture did not convert at two months | 158 (36.8) | 122 (25.6) | 65 (70.6) |
| Smear did not convert at two months | 110 (25.6) | 90 (19.1) | 50 (54.3) |
| Derivation group | 343 (60) | 285 (59.6) | 58 (63.0) |

*SD: Standard deviation

**Table 3. Demographic and clinical characteristics of patients with multidrug-resistant tuberculosis in the derivation and validation groups, from Hunan Chest and Gondar University Hospitals, 2010–2014.**

| Risk factors | Derivation (N = 227) | Validation (N = 343) |
|---|---|---|
| | Mean (SD*) | Mean (SD*) |
| Age (years) | 39.7± 13.7 | 38.5 ± 12.5 |
| | Number (%) | Number (%) |
| Male sex | 157 (69.2) | 239 (69.7) |
| Farmer or daily labourer | 183 (80.6) | 268 (78.1) |
| History of TB treatment | 214 (94.3) | 323 (94.2) |
| History of 2nd line TB drug treatment | 59 (25.9) | 79 (23.0) |
| Resistance to ethambutol | 83 (36.5) | 101 (29.4) |
| Resistance to any injectable TB drugs | 131 (57.7) | 178 (51.9) |
| Resistance to any fluoroquinolones | 17 (7.5) | 29 (8.4) |
| Culture did not convert at two months | 75 (33.2) | 112 (32.6) |
| Smear did not convert at two months | 54 (24.3) | 86 (25.2) |

*SD: Standard deviation

**Table 4. Crude hazard ratios from the univariable Cox proportional hazard model in the derivation group in patients with multidrug-resistant tuberculosis, from Hunan Chest and Gondar University Hospitals, 2010–2014.**

| Risk factors | Crude Hazard ratio (95% CI) | P value |
|---|---|---|
| Mean age | 1.0 (0.9–1.0) | 0.81 |
| Male sex | 1.3 (0.7–2.4) | 0.37 |
| Farmer and daily labourer | 0.6 (0.3–1.2) | 0.15 |
| History of TB treatment | 3.3 (0.5–24.2) | 0.23 |
| **History of 2nd line TB drug treatment** | **1.7 (1.1–3.0)** | **0.04** |
| Resistance to ethambutol | 1.5 (0.9–2.7) | 0.11 |
| Resistance to any injectable TB drugs | 1.7 (0.9–3.0) | 0.06 |
| **Resistance to any fluoroquinolones** | **3.1 (1.6–5.8)** | **0.001** |
| **Culture did not convert at two months** | **3.4 (1.9–5.9)** | **0.000** |
| **Smear did not convert at two months** | **2.8 (1.6–4.8)** | **0.000** |

CI: confidence interval

In the derivation group, the proportion of patients with a poor treatment outcome was 14% in the low-risk group, 27% in the medium-risk group, and 71% in the high-risk group. In the validation group, the corresponding poor treatment outcome rates were 14%, 18%, and 54%, respectively (Fig 2).

**Table 5. Adjusted hazard ratios, regression coefficients (β), and point score from the multivariable Cox proportional hazard model in the derivation group in patients with multidrug-resistant tuberculosis, from Hunan Chest and Gondar University Hospitals, 2010–2014.**

| Risk factors | Adjusted Hazard Ratio (95% CI) | β-coefficient | P value | Points |
|---|---|---|---|---|
| History of second-line TB treatment | 1.7 (1.1–3.2) | 0.5 | 0.04 | 2 |
| Resistance to fluoroquinolones | 2.5 (1.3–4.9) | 0.9 | 0.007 | 3 |
| Smear did not convert at two months | 3.0 (1.7–5.3) | 1.1 | <0.001 | 4 |

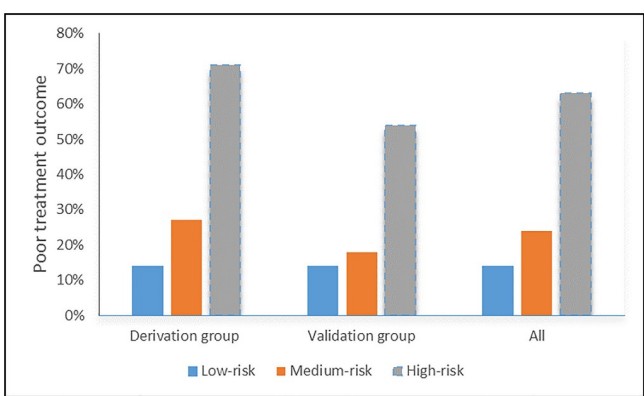

**Fig 2. The proportion of patients experiencing a poor treatment outcome by clinical score in the derivation group, the validation group and for all patients.**

## Discriminatory performance of the risk score

Discrimination measures of the derivation and validation groups are shown in Fig 3. The risk score developed in this study had a moderate degree of accuracy to discriminate between poor and successful treatment outcomes. The area under the ROC curve for the point system in the derivation group was 0.69 (95% CI: 0.60 to 0.77) and was similar to that in the validation group (0.67; 95% CI: 0.56 to 0.78; p = 0.82).

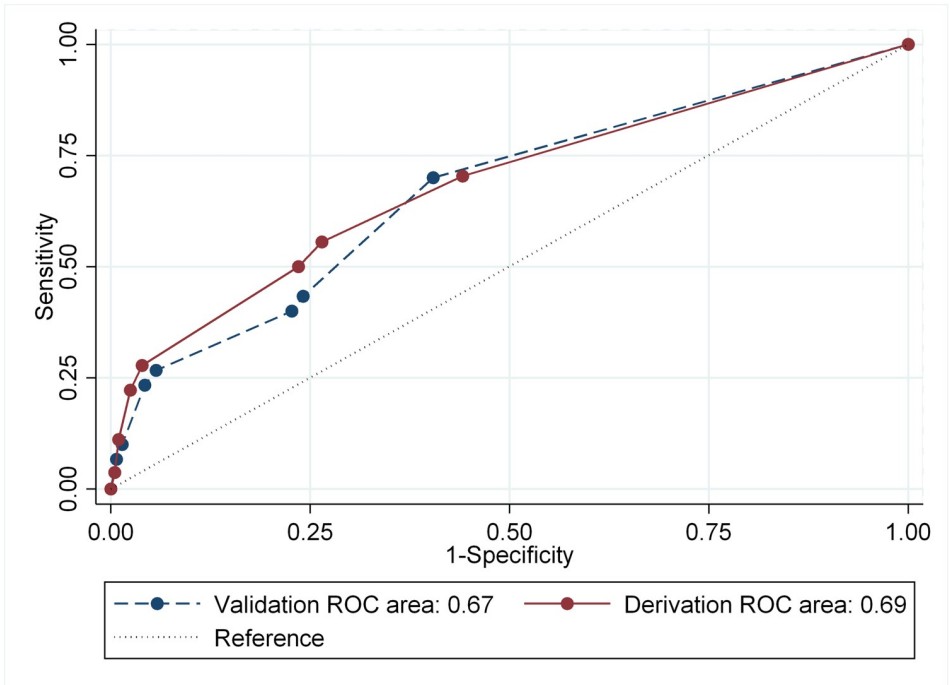

**Fig 3. Receiver operating characteristic curve analysis for the derivation and validation groups in patients with multidrug-resistant tuberculosis.**

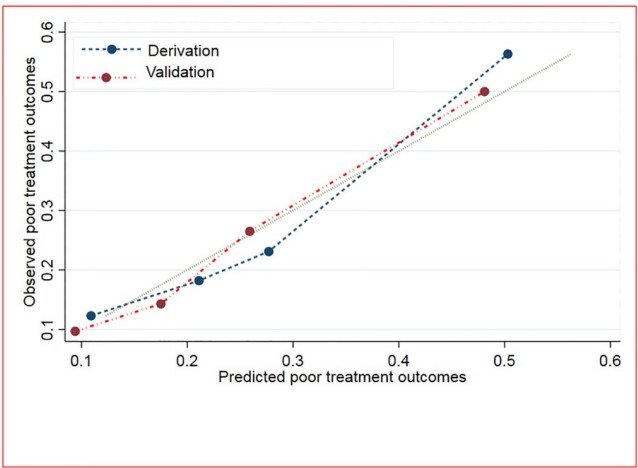

**Fig 4. Calibration of the model using predicted and observed probabilities of poor treatment outcome for both derivation and validation groups among patients with multidrug-resistant tuberculosis.**

### Calibration of the risk scores

The calibration of the risk score is presented graphically by plotting the observed and predicted poor treatment outcomes. Overall, the model displayed modest calibration performance in the derivation and validation groups (Fig 4).

## Discussion

The risk score developed in this study has a moderate degree of accuracy to discriminate between poor and successful treatment outcomes, with an AUROC score of 0.69. This discriminatory performance is higher than other risk scores which have been developed for coronary heart diseases (AUROC = 0.58–0.63) [16, 17]. However, the accuracy of the risk score in our study is also lower than risk scores which have been developed to predict the risk of severe dengue (0.90) [15], paediatric mortality (0.88) [18], pulmonary embolism [19] (0.85), death from Chagas' Heart Disease (0.81) [27], dementia (0.77) [28], and atrial fibrillation (0.78) [29]. The reason for the lower discriminatory performance in our study as compared to the studies described above could be due to the fact that we used secondary data available from the MDR-TB register, and as this dataset is limited, some confounding variables might be missing and data quality might not be optimal. In contrast, previous studies developed their risk score using primary data collected from study participants. However, developing a risk score using routinely collected data by national TB programmes has an advantage because it might be easily applied to all patients without any additional costs.

Our study showed that clinical risk factors such as resistance to fluoroquinolone antibiotics, history of taking second-line TB treatment, and no smear conversion at two months predicted poor treatment outcomes in patients with MDR-TB. A risk score developed by adding points from each of these variables classified patients into categories of low, medium and high-risk for a poor treatment outcome with a moderate degree of accuracy. Patients in the high-risk group could potentially benefit from intensive treatment, including psychological support, financial assistance, and regular follow-up [30–33].

Although a clinical risk score has not been yet developed for MDR-TB, the factors that are included in our scoring system have been identified as predictors of a poor treatment outcome for MDR-TB patients in previous studies [34–37]. A systematic review and meta-analysis, which was conducted to assess the impact of fluoroquinolones resistance on treatment

outcomes, showed that resistance to fluoroquinolones increases the risk of treatment failure, death and relapse in patients with MDR-TB [38]. Another study conducted in Estonia showed that history of previous TB treatment is a predictor of poor treatment outcomes in patients with MDR-TB [37]. A study conducted in China showed that sputum smear conversion (from positive to negative) at two months after treatment has commenced is a potential predictor of MDR-TB treatment outcome [36].

Our study showed that resistance to fluoroquinolone antibiotics predicted poor treatment outcomes in patients with MDR-TB. Fluoroquinolones (which included levofloxacin, moxifloxacin, and gatifloxacin) are broad-spectrum antibiotics and are widely used to treat several bacterial infections [39, 40]. They are an essential component of an MDR-TB treatment regimen [41]. WHO recommends that fluoroquinolones are considered to be the most important component of the core MDR-TB regimen and they should always be included unless there is evidence for absolute contraindication for their use [42]. Resistance to fluoroquinolones, which predominantly occurs due to prior exposure [41], continues to rise globally and is a major clinical problem [43, 44]. The 2017 Global Tuberculosis Report reported a fluoroquinolone resistance rate of 20% in MDR-TB strains tested [7]. The detection of resistance to fluoroquinolones (using line probe assays) is included in the End-TB Strategy as part of calls for universal access to drug susceptibility testing (DST) [45]. Tuberculosis patients with fluoroquinolones resistance may require regimen modification [46], such as the incorporation of bedaquiline into the regimens [47].

Identifying groups at high risk for a poor treatment outcome, and providing appropriate treatment would be cost-effective, especially in resource limited settings. Treatment of MDR-TB is complex [4], and requires combinations of second-line TB drugs which are less effective and more toxic than first-line TB drugs [2]. To increase the probability of treatment success, high-risk patients (e.g. those with a risk score > 6 points) may require resource-intensive support such as the provision of financial support, psychological counselling, and regimen adjustment [32, 33]. Thus, our scoring system may be useful for clinicians to identify high-risk groups in order to provide additional financial and psychosocial support for patients during MDR-TB treatment.

This is, to our knowledge, the first published report on the development of a simple scoring system to predict poor treatment outcomes (i.e. death, lost to follow up or treatment failure) in patients with MDR-TB. The scoring system is based on variables that are commonly recorded for surveillance purposes and in clinical practice. However, the study has several limitations. First, since our study is limited by the retrospective nature of the data, some variables which we did not include in our model might have contributed to the prediction of poor treatment outcomes. Thus, further studies are needed to develop a scoring system which also takes into account the potential risk factors. Second, we included all deaths occurring during the course of treatment; however, some deaths may not have been related to MDR-TB. Third, it would be more meaningful if we could calculate the risk score using data from one of the study hospitals and validate this score using the data obtained from the other hospital. However, due to the fact that we had very few data from University of Gondar Hospital, it was impractical to do this. Thus, we preferred to merge the data and randomly divide it into derivation and validation groups. Therefore, our score system requires external validation using larger datasets and through the use of prospective data in other settings or countries.

## Conclusions

This new risk score can be used to estimate the absolute risk of poor treatment outcome in patients with MDR-TB. Resistance to any fluoroquinolones, history of previous second-line

TB treatment, and smear conversion status could help to identify high-risk MDR-TB patients for poor treatment outcomes. However, our scoring system should be externally validated and further improved to increase its predictive value.

## Supporting information

**S1 File. Data supporting the findings of this study.**
(ZIP)

## Acknowledgments

The authors would like to thank those who participated in the process of data extraction from the medical records.

## Author Contributions

**Conceptualization:** Kefyalew Addis Alene.

**Data curation:** Kefyalew Addis Alene, Zuhui Xu.

**Formal analysis:** Kefyalew Addis Alene.

**Investigation:** Kefyalew Addis Alene.

**Methodology:** Kefyalew Addis Alene.

**Supervision:** Kerri Viney, Darren J. Gray, Emma S. McBryde, Archie C. A. Clements.

**Validation:** Kefyalew Addis Alene.

**Writing – original draft:** Kefyalew Addis Alene.

**Writing – review & editing:** Kerri Viney, Darren J. Gray, Emma S. McBryde, Zuhui Xu, Archie C. A. Clements.

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
