## [Decision Letter · Decision Letter 0]

5 Nov 2019

PONE-D-19-25758

Development of a risk score for prediction of poor treatment outcomes among patients with multidrug-resistant tuberculosis

PLOS ONE

Dear Dr. Alene,

Thank you for submitting your manuscript to PLOS ONE. After careful consideration, we feel that it has merit but does not fully meet PLOS ONE’s publication criteria as it currently stands. Therefore, we invite you to submit a revised version of the manuscript that addresses the points raised during the review process.

We would appreciate receiving your revised manuscript by Dec 20 2019 11:59PM. To enhance the reproducibility of your results, we recommend that if applicable you deposit your laboratory protocols in protocols.io, where a protocol can be assigned its own identifier (DOI) such that it can be cited independently in the future. For instructions see: http://journals.plos.org/plosone/s/submission-guidelines#loc-laboratory-protocols

We look forward to receiving your revised manuscript.

Kind regards,

HASNAIN SEYED EHTESHAM

Academic Editor

PLOS ONE

Journal Requirements:

2. In your ethics statement in the manuscript and in the online submission form, please provide additional information about the patient records used in your retrospective study. Specifically, please ensure that you have discussed whether all data were fully anonymized before you accessed them and/or whether the IRB or ethics committee waived the requirement for informed consent.

Additional Editor Comments:

This manuscript has been reviewed by 2 reviewers. Both reviewers feel that this manuscript should be accepted. I have also gone through this manuscript and also the comments of both the reviewers. I recommend this manuscript for a minor revision.

Reviewers' comments:

Reviewer's Responses to Questions

**Comments to the Author**

1. Is the manuscript technically sound, and do the data support the conclusions?

Reviewer #1: Yes

Reviewer #2: Partly

2. Has the statistical analysis been performed appropriately and rigorously? 

Reviewer #1: Yes

Reviewer #2: Yes

3. Have the authors made all data underlying the findings in their manuscript fully available?

Reviewer #1: Yes

Reviewer #2: Yes

4. Is the manuscript presented in an intelligible fashion and written in standard English?

Reviewer #1: Yes

Reviewer #2: Yes

5. Review Comments to the Author

Reviewer #1: Reviewer #: It is good research article about risk score development for prediction of poor treatment outcomes among patients with multidrug-resistant tuberculosis (MDR-TB) using routinely collected data from two large countries that have a high MDR-TB burden and are geographically, economically and epidemiologically distinct.

Further, this is the first published report on the development of a simple scoring system to predict poor treatment outcomes (death, lost to follow up or treatment failure) in patients with MDR-TB.

The authors introduced the statistical method and analysis of the poor outcomes among patients with multidrug-resistant tuberculosis, which is helpful to better understand the current methods about detection and treatment failure in patients with MDR-TB.

This developed risk score can be useful to identify and estimate the absolute risk of poor treatment outcome in patients with MDR-TB.

Thus, I suggest acceptance of this article, as this study and risk score can be helpful for the clinicians to provide a better medical care and financial and psychosocial support to patients undergoing MDR-TB treatment.

Reviewer #2: Authors of the manuscript entitled "Development of a risk score for prediction of poor treatment outcomes among patients

with multidrug-resistant tuberculosis" have attempted to develop a statistical model aimed at deciphering risk scores for poor treatment outcomes among patients with multidrug resistance TB. Though similar studies based on risk scores for prediction of poor treatment outcome in the case of drug susceptible TB, dengue, coronary heart diseases, pulmonary embolism etc. have been previously reported but not in the case of MDR-TB.

This manuscript can be accepted with following minor revisions that may add further value to the work.

1>A detailed study flow chart may be included for clarity.

2>Lines 164-170: Authors preferred use of smear conversion rates over culture conversion rates for risk score prediction. Sensitivity of smear conversions analyses may not be one of the good variable for risk score predictions thus leading to a moderate AUROC of 0.69 that is much less than the 0.9 and might compromise the applicability of the risk scores for prediction of poor treatment outcomes. Thus it may be prudent to include clinical factors with high sensitivity and specificity for prediction of risk scores and thus an improved AUROC.

3>Authors have merged data from patient cohorts of two geographically distinct countries with high MDR-TB burden. Though it would have been of high significance if they had carried out risk score based predictions on individual cohorts using statistically significant numbers.

4> This study may be a good attempt toward carrying out risk score based predictions of poor treatment outcome in the case of MDR-TB patients and may thus help clinicians in decision making in the case of MDR-TB patients.

6. PLOS authors have the option to publish the peer review history of their article (what does this mean?). If published, this will include your full peer review and any attached files.

Reviewer #1: No

Reviewer #2: No

---

## [Author Response · Author response to Decision Letter 0]

27 Nov 2019

Response to editor and reviewers’ comment

Ref.: PONE-D-19-25758

Title: Development of a risk score for prediction of poor treatment outcomes among patients with multidrug-resistant tuberculosis

Journal: PLOS ONE

Authors: Kefyalew Addis Alene, Kerri Viney, Darren J Gray, Emma S McBryde, Zuhui Xu, Archie CA Clements

Dear Editor Prof Hasnain Seyed Ehtesham

Subject: Submission of revised paper mentioned above 

Thank you for the opportunity to revise our manuscript. We appreciate the thoughtful review and constructive suggestions. We have carefully reviewed the comments and have revised the manuscript accordingly. Following this letter are the editor and reviewer comments with our point-by-point responses for each specific comment raised by the reviewers in green font, including how and where the text was modified. The change made in the manuscript is shown in a separate track change document and uploaded to the system with the revised version of the manuscript. We hope the revised version is now suitable for publication and look forward to hearing from you in due course. 

Yours sincerely, 

Kefyalew Addis Alene 

Corresponding Author 

Response to editor’s comment 

1. When submitting your revision, we need you to address these additional requirements. Please ensure that your manuscript meets PLOS ONE's style requirements, including those for file naming. 

Response: We followed the PLOS ONE style templates when submitting the revised version of the manuscript. 

2. In your ethics statement in the manuscript and in the online submission form, please provide additional information about the patient records used in your retrospective study. Specifically, please ensure that you have discussed whether all data were fully anonymized before you accessed them and/or whether the IRB or ethics committee waived the requirement for informed consent.

Response: We have now mentioned that “Ethics approval was obtained from the Australian National University Human Research Ethics Committee (protocol number 2016/218) and from the Institutional Review Board of the University of Gondar. Permission was granted to access the secondary data from Tuberculosis Control Institute of Hunan Province, and this was documented in a letter. All data were fully anonymized before we accessed them, and the study was conducted in collaboration with researchers from Gondar University and Hunan Chest Hospital.”

 Additional Editor Comments

3. This manuscript has been reviewed by 2 reviewers. Both reviewers feel that this manuscript should be accepted. I have also gone through this manuscript and also the comments of both the reviewers. I recommend this manuscript for a minor revision.

Response: We greatly appreciate the editor for giving us a chance to revise our manuscript. We have carefully reviewed the comments and have revised the manuscript accordingly.

Response to reviewers’ comment 

Reviewer #1: It is good research article about risk score development for prediction of poor treatment outcomes among patients with multidrug-resistant tuberculosis (MDR-TB) using routinely collected data from two large countries that have a high MDR-TB burden and are geographically, economically and epidemiologically distinct. Further, this is the first published report on the development of a simple scoring system to predict poor treatment outcomes (death, lost to follow up or treatment failure) in patients with MDR-TB. The authors introduced the statistical method and analysis of the poor outcomes among patients with multidrug-resistant tuberculosis, which is helpful to better understand the current methods about detection and treatment failure in patients with MDR-TB. This developed risk score can be useful to identify and estimate the absolute risk of poor treatment outcome in patients with MDR-TB. Thus, I suggest acceptance of this article, as this study and risk score can be helpful for the clinicians to provide a better medical care and financial and psychosocial support to patients undergoing MDR-TB treatment.

Response: We greatly appreciate the thoughtful comments of the reviewer.

Reviewer #2: Authors of the manuscript entitled "Development of a risk score for prediction of poor treatment outcomes among patients with multidrug-resistant tuberculosis" have attempted to develop a statistical model aimed at deciphering risk scores for poor treatment outcomes among patients with multidrug resistance TB. Though similar studies based on risk scores for prediction of poor treatment outcome in the case of drug susceptible TB, dengue, coronary heart diseases, pulmonary embolism etc. have been previously reported but not in the case of MDR-TB. This manuscript can be accepted with following minor revisions that may add further value to the work.

1. A detailed study flow chart may be included for clarity.

Response: We now include the following flow chart in the revised version of the manuscript for clarity

 721 patients with drug resistant TB in two hospitals 

120 not bacteriologically confirmed 

 601 bacteriologically confirmed patients 

22 did not start MDR-TB treatment 

 579 MDR-TB patients who started treatment 

9 treatment outcomes were not recorded 

 570 MDR-TB patients with recorded treatment outcomes 

Figure 1: Flowchart of eligible participants for our study on the development of a risk score for prediction of poor treatment outcomes among patients with multidrug-resistant tuberculosis, and reasons for exclusion. 

2. Lines 164-170: Authors preferred use of smear conversion rates over culture conversion rates for risk score prediction. Sensitivity of smear conversions analyses may not be one of the good variable for risk score predictions thus leading to a moderate AUROC of 0.69 that is much less than the 0.9 and might compromise the applicability of the risk scores for prediction of poor treatment outcomes. Thus it may be prudent to include clinical factors with high sensitivity and specificity for prediction of risk scores and thus an improved AUROC.

Response: When we checked for the presence of multi-collinearity, a high degree of collinearity was observed between smear and culture conversion status. Thus, we selected smear conversion for the final model to avoid the observed multi-collinearity. We selected smear conversion because it is less expensive, widely available, and more convenient than culture conversion to predict the treatment outcomes of patients at peripheral and lower-resource settings. However, we run two different models that contain culture conversion and smear conversion a sensitivity analysis, and we found similar AUROC results. 

3. Authors have merged data from patient cohorts of two geographically distinct countries with high MDR-TB burden. Though it would have been of high significance if they had carried out risk score-based predictions on individual cohorts using statistically significant numbers.

Response: As we have mentioned in the limitation section of the manuscript, it would be more meaningful if we could calculate the risk score using data from one of the study hospitals and validate this score using the data obtained from the other hospital. However, since we had very few data from University of Gondar Hospital, it was impractical to do this. Thus, we preferred to merge the data and randomly divide it into derivation and validation groups. Therefore, our score system requires external validation using larger datasets and using prospective data in other settings or countries. 

4. This study may be a good attempt toward carrying out risk score-based predictions of poor treatment outcome in the case of MDR-TB patients and may thus help clinicians in decision making in the case of MDR-TB patients. 

Response: We appreciate the thoughtful comments of the reviewer.

---

## [Editor Report · Decision Letter 1]

13 Dec 2019

Development of a risk score for prediction of poor treatment outcomes among patients with multidrug-resistant tuberculosis

PONE-D-19-25758R1

Dear Dr. Alene,

We are pleased to inform you that your manuscript has been judged scientifically suitable for publication and will be formally accepted for publication once it complies with all outstanding technical requirements.

With kind regards,

HASNAIN SEYED EHTESHAM

Academic Editor

PLOS ONE

Additional Editor Comments (optional):

I have gone through the revised manuscript and also the Authors response to reviewers comment. The authors have satisfactorily addressed all the comments made by the reviewers and have revised the manuscript accordingly.

I recommend this manuscript for publication.
---

## [Editor Report · Acceptance letter]

20 Dec 2019

PONE-D-19-25758R1 

Development of a risk score for prediction of poor treatment outcomes among patients with multidrug-resistant tuberculosis 

Dear Dr. Alene:

I am pleased to inform you that your manuscript has been deemed suitable for publication in PLOS ONE. Congratulations! Your manuscript is now with our production department. 

With kind regards,

on behalf of

Prof HASNAIN SEYED EHTESHAM 

Academic Editor

PLOS ONE